# Sleep Continuity, Stability and Cyclic Organization Are Impaired in Insomniacs: A Case–Control Study

**DOI:** 10.3390/ijerph20021240

**Published:** 2023-01-10

**Authors:** Francesca Conte, Serena Malloggi, Oreste De Rosa, Ilaria Di Iorio, Federica Romano, Fiorenza Giganti, Gianluca Ficca

**Affiliations:** 1Department of Psychology, University of Campania L. Vanvitelli, Viale Ellittico 31, 81100 Caserta, Italy; 2Department NEUROFARBA, University of Firenze, Via di San Salvi 12, 50135 Firenze, Italy

**Keywords:** insomnia, sleep quality, sleep continuity, sleep stability, sleep cycles

## Abstract

The possibility of distinguishing insomniacs from good sleepers based on polysomnography (PSG) remains an open question. While these groups show modest differences in traditional PSG parameters, some studies suggest that finer measures may be more useful. Here we assess differences between good sleepers (GS), poor sleepers (PS) and insomniacs (IN) in classical PSG measures as well as in sleep continuity, stability and cyclic organization. PSG-monitored sleep (two nights) of 17 IN (diagnosed through a standard clinical interview; Pittsburgh Sleep Quality Index (PSQI) ≥ 5, Insomnia Severity Index (ISI) > 14) was compared to that of 33 GS (PSQI < 5) and 20 PS (PSQI ≥ 5, ISI ≤ 14). Compared to GS, IN were impaired in sleep macrostructure (sleep latency, sleep efficiency, WASO%) and in continuity, stability and organization, whereas PS only showed disrupted continuity and stability. Spindle parameters were comparable between IN and GS, but the former displayed enhanced power in fast frequency bands. Our findings support the hypothesis of a continuum between individuals with self-reported poor sleep and insomniacs. Further, they add to extant data on impaired sleep continuity, stability and organization in poor sleepers and elderly individuals, underlining the utility of including these measures in standard sleep assessments.

## 1. Introduction

Insomnia is a very common and debilitating sleep disorder, affecting about 10% of the general population [1]. Unsurprisingly, a significant amount of research has been produced with the attempt to define its causes and characteristics, as well as to propose increasingly effective management strategies. However, as highlighted by several authors [2,3], an important caveat of literature on insomnia is the lack of standardization of the protocols used for evaluating and defining insomnia symptoms, resulting in a great variability in the selection of insomniac participants across the different studies. 

The publication of research diagnostic criteria for insomnia [4] and of consensus recommendations on standard assessment procedures and measures to be used in insomnia-focused research [2] represents a relevant effort produced to reduce this limitation. Along the same line, some studies have proposed quantitative cut-offs derived from sleep diaries that may be usefully employed as additional selection criteria for the screening of insomniac samples, e.g., [5]. 

In parallel, other studies have attempted to use PSG measures as selection criteria, often with the aim of verifying whether a certain intervention was able to improve specific sleep variables (e.g., [6]). However, many individuals diagnosed with insomnia do not show objective sleep disruptions during PSG monitoring (see [7] for a review). Furthermore, the sensitivity and specificity of common sleep parameters (i.e., sleep onset latency, total sleep time, sleep efficiency, wake after sleep onset), as well as of several previously proposed PSG criteria sets, have been found to be insufficient for the discrimination of insomniacs from good sleepers [3]. 

This literature is actually in line with the available data on the relationships between subjective and objective sleep quality. Indeed, discrepancies between the two are also reported in healthy populations (e.g., [8,9]), and there is still much debate on the objective determinants of subjective sleep quality ratings, with studies pointing to a number of different parameters as the main factors determining sleep quality perception (e.g., [10,11,12]). 

In this context, in line with other authors [13,14], we have previously argued that traditional sleep measures (e.g., sleep efficiency, total sleep time, sleep stage proportions) could be insufficient to detect the subtle impairments of the sleep episode and more fine-grained analyses could be necessary to properly describe objective sleep quality [15,16]. Specifically, we observed that more specific measures of sleep fragmentation (e.g., frequency of long and brief awakenings), as well as of sleep instability (e.g., frequency of arousals and state transitions) and disorganization (e.g., number of complete sleep cycles, time spent in sleep cycles) differed between the sleep episodes of young and elderly subjects [15] and of self-reported good and poor sleepers [16]. Moreover, we showed that improvements in these parameters after bedtime cognitive training (not accompanied by corresponding improvements in traditional sleep measures) were reflected in better sleep quality perception in poor sleepers [17]. In addition, we observed a high degree of sleep fragmentation (actigraphically detected) in the face of preserved global sleep architecture and duration during the quarantine imposed in Italy in response to the third COVID-19 pandemic wave [18], suggesting that the significant subjective sleep disruptions reported during the pandemic (e.g., [19,20]) could have relied on these subtle sleep disruptions rather than gross sleep structure alterations. Overall, these data suggest that impairments in these kinds of sleep features could affect sleep quality ratings, thus contributing, when undetected, to the dissociation between subjective and objective sleep measures. 

This reasoning could also be applied to insomniac populations. In other words, as previously suggested [21,22], it is plausible that the objective determinants of sleep quality perception rely in sleep features which are not easily (or not at all) accessible to conscious awareness, producing the apparent objective–subjective discrepancies so often observed in insomniacs. Data in support of this idea come from a few studies showing, in insomniacs relative to good sleepers, a higher cyclic alternating pattern rate (e.g., [23]) and more elevated activities in rapid frequency bands (e.g., [24,25]). 

Using a case–control study design, here we assess PSG-defined sleep parameters in a sample of insomnia sufferers compared to a control group of good sleepers, with a special focus on a set of measures indexing sleep continuity, stability and cyclic organization that have not been previously assessed in this population. Specifically, sleep continuity is usually addressed through sleep efficiency and wake after sleep onset time, whereas we also consider the frequency of brief and long awakenings, as well as their average duration. Similarly, we measure sleep stability through the frequency of arousals, microarousal index, and periods of functional uncertainty (i.e., sleep periods in which the frequency of stage transitions is particularly high [15]), in addition to the frequency of sleep stage transitions. Finally, we analyze a set of variables indexing sleep cyclic organization, which, to our knowledge, has never been described in insomniac populations. As mentioned above, we recently observed that this set of measures is impaired in healthy poor sleepers (i.e., individuals with subjective reports of poor sleep who do not satisfy criteria for an insomnia disorder) compared to good sleepers [16]. Considering that the development of insomnia is a frequent long-term evolution of perceived poor sleep quality [26,27], we hypothesize that insomniac participants will show higher sleep fragmentation, instability and disorganization relative to controls. From this perspective, we also compare sleep measures of our insomniac sample to those of the group of poor sleepers from our previous study [16]. 

## 2. Materials and Methods

### 2.1. Participants

Eighty-two potential participants were approached at university sites (i.e., lecture halls, library, etc. of the University of Campania “L. Vanvitelli”) and asked to fill out a set of screening questionnaires: the Pittsburgh Sleep Quality Index (PSQI) [28], the Insomnia Severity Index (ISI) [29], the Sleep Disorder Questionnaire (SDQ) [30], the Beck Depression Inventory II (BDI-II) [31], and the Beck Anxiety Inventory (BAI) [31]. In addition, participants underwent a semi-structured interview at the sleep laboratory of the Department of Psychology (University of Campania “L. Vanvitelli”), conducted by a licensed psychologist who had received specific training, in order to assess general medical condition and health habits and the presence of psychiatric disorders and of sleep disorders. The presence of clinical insomnia was specifically addressed by means of the standard semi-structured interview [32].

Based on scores at the screening instruments and on the interview, 50 university students were recruited for the study, according to the following inclusion criteria: absence of any relevant somatic or psychiatric disorder; absence of moderate or severe depression and anxiety symptoms (BDI-II score < 19 [33], BAI score < 16 [34]); no history of drug or alcohol abuse; absence of sleep disorders (other than insomnia for the IN group) and of any sleep apnea or respiratory disorder symptom; having a regular sleep–wake pattern (e.g., individuals with irregular study or working habits, such as shift-working, were excluded); and no use of psychoactive medication or alcohol at bedtime. Thirty-two potential participants were instead excluded for the following reasons: presence of moderate or severe depression and/or anxiety symptoms (n = 9), history of drug use (n = 4), regular use of alcohol in the evening (n = 7), having an irregular sleep–wake pattern (n = 9), and use of psychoactive medication (n = 3). 

Of the 50 recruited participants, 33 were included in the “Good Sleep group” (GS: 12M, 21F, age = 27.3 ± 6.18) and 17 in the “Insomnia group” (IN: 4M, 13F, age = 26.6 ± 6.71). In addition to the inclusion criteria common to both groups, inclusion in the GS group was based on: PSQI score < 5 (i.e., cut-off for subjective poor sleep [35]), ISI score < 8 (absence of insomnia [36]), being classified as “good sleeper” at the SDQ, and absence of any sleep disorder as also verified through the interview. Instead, for inclusion in the IN group, participants had to score ≥5 at the PSQI, >14 at the ISI (i.e., presence of moderate or severe insomnia [36]) and to be classified as presenting “clinically significant insomnia” at the SDQ; furthermore, they had to fully meet DSM-5 TR criteria for Insomnia Disorder [37], as verified through the interview. 

The IN and GS groups did not differ in circadian preference, measured through the reduced version of the Morningness–Eveningness Questionnaire [38] (GS: 13.6 ± 2.56, IN: 14.4 ± 3.58, Student’s t = 1.12, *p* = 0.271) or habitual sleepiness levels, measured through the Epworth Sleepiness Scale [39] (GS: 6.47 ± 3.58, IN: 6.71 ± 3.60, Student’s t = −0.18, *p* = 0.852). Depression and anxiety levels were higher in IN (BDI-II: 12.16 ± 7.22; BAI: 11.2 ± 6.5) than GS participants (BDI-II: 5.57 ± 3.76; BAI: 3.36 ± 2.95) (BDI-II: χ^2^ = 5.46, *p* = 0.020; BAI χ^2^ = 9.01, *p* = 0.003).

The sleep parameters of the IN and GS groups were compared, in this study, with those recorded over two nights in a sample of healthy poor sleepers in a previous study from our group [16]. In that study, the poor sleepers (PS) were selected as individuals with subjective reports of poor sleep but not satisfying the criteria for an insomnia disorder. Specifically, they had to adhere to the same general inclusion criteria as those employed here for GS and IN subjects (verified through an ad hoc interview, the BAI and the BDI-II): absence of any relevant somatic or psychiatric disorder; absence of moderate or severe depression and anxiety symptoms (BDI-II score < 19 [33], BAI score < 16 [34]); no history of drug or alcohol abuse; absence of sleep disorders and of any sleep apnea or respiratory disorder symptom; having a regular sleep–wake pattern (e.g., individuals with irregular study or working habits, such as shift-working, were excluded); no use of psychoactive medication or alcohol at bedtime. In addition, participants had to be classified as “poor sleepers” (score ≥ 5) at the PSQI [35] and to score ≤ 14 at the ISI (i.e., absence of clinically significant insomnia [36]). The sample consisted of 20 PS (4M, 16F; age: 24.5 ± 2.34 years; PSQI score range: 5–14). Details on recruitment and procedures used in that study are described in detail in [16]. As displayed in Table 1, demographic factors did not differ across the three groups (GS, IN and PS). 

Recruitment and data collection for GS and IN participants took place between February and June 2022. Data on the PS participants (published in 2021 [16]) were obtained in the second half of 2019 at the University of Campania “L. Vanvitelli”. The recruitment process is schematized in Figure 1.

The study design was preliminarily submitted to the Ethical Committee of the Department of Psychology, University of Campania “L. Vanvitelli”, which approved the research (code 22/2020) and certified that the involvement of human participants was performed according to acceptable standards. All participants signed a consent form prior to enrollment in the study. They received no money or credit compensation for their participation.

### 2.2. Procedure

In order to minimize confounds linked to experimental settings and to preserve ecological validity, the procedure was conducted at the participants’ homes. After an adaptation night, each subject of the IN and GS groups underwent two nights of sleep recording at home, with 9 h time in bed (TIB) allotted. Recording nights were scheduled with intervals of one week.

For the 3 days preceding each sleep recording session, subjects were requested to maintain regular bedtimes and rising times and regular napping habits (they were allowed to take naps only if these represented a daily habit: in that case, naps should not differ from the subject’s usual naps either in length or in circadian placement). To control for potential effects of daily cognitive activities (reviewed in [40]), participants were also instructed to keep daily activities as habitual as possible and to avoid, on recording days, any cognitively engaging activity (such as reading, studying, playing cards, etc.) beyond habit. These factors were checked by requiring subjects to fill in a detailed sleep log and a short ad hoc diary on daily activities. On recording days, the experimenter arrived at the subject’s house approximately 40 min before usual bedtime and proceeded to electrode montage. Subjects were instructed to go to bed immediately after that.

### 2.3. Sleep Recordings and Analysis

Polysomnographic recordings were performed by recording six electroencephalographic (EEG channels: F3, F4, C3, C4, O1, O2, referenced against contralateral mastoids A1 and A2), two electro-oculographic (LOC-A2, ROC-A1), and a bipolar submental electromyogram channels, according to standard guidelines [41]. Data were acquired by means of a BluNet multichannel recording system (Ne.Ro SRL, Florence, Italy) at a sample rate of 200 Hz. Sleep recordings were band-passed (0.3–35 Hz) and then visually scored according to standard criteria [41] by an expert technician, blind to the study groups. To verify scoring reliability, 10 randomly selected sleep recordings were scored by another technician. Inter-rater agreement was 92%.

Classical architecture sleep variables considered in the study were: time in bed (TIB; i.e., time from lights off to final awakening, in minutes), total sleep time (TST; i.e., time from the first appearance of N1 to final awakening, in minutes), actual sleep time (AST; i.e., time spent in sleep states, in minutes), sleep onset latency in minutes (SOL), sleep stage proportions over TST (N1%, N2%, N3%, REM%), percentage of wake after sleep onset over TST (WASO%) and sleep efficiency (SE%; i.e., percentage of AST over TIB).

Moreover, objective sleep quality was evaluated through an additional set of variables (as in [16]), indexing:sleep continuity: total awakenings frequency per hour of AST; frequency of brief (<4 epochs) and long (≥4 epochs) awakenings per hour of AST; frequency of awakenings from N1, N2, N3 and REM per minute of that stage;sleep stability: frequency of arousals per hour of AST (arousals were defined as all transitions to shallower NREM sleep stages and from REM sleep to N1); frequency of state transitions per hour of TST (state transitions were defined as all transitions from one state to another, including all those to and from wake, and all those from one stage to another); frequency of “Functional Uncertainty Periods” (“FU periods”) per hour of TST (FU periods were defined as periods in which a minimum of three state transitions follow one another with no longer than 90-sec intervals, e.g., in a sequence of 10 epochs scored as follows: N2-N3-N3-N2-N2-N2-N3-N3-REM-REM-REM-REM, the first 8 epochs would be labelled as an FU period, but not the last 4, since the REM state lasts more than 90 s); percentage of total time spent in FU over TST (TFU%); frequency of arousals from N2 to N1, from N3 to N1, from N3 to N2 and from REM to N1 per minute of that stage;sleep organization: number of complete sleep cycles, defined as the sequences of NREM and REM sleep (each lasting at least 10 min) not interrupted by wake periods longer than 2 min; proportion of total time spent in cycles over TST (TCT%); and mean duration of cycles, in minutes.

Microarousals were detected by means of an automatic analysis performed through the Polysmith software package (Nihon Kohden Polysmith version 9.0) and based on the following definition [42]: an abrupt change in EEG frequency, including theta, alpha and/or frequencies >16 Hz (but not spindles), lasting from 3 to 14 s. From this analysis we derived the microarousal index (n/AST h). 

Sleep spindles (total spindles: 11–16 Hz; slow spindles: 9–12.5 Hz; fast spindles: 12.5–16 Hz) were automatically detected using the same software from the central derivations (C3 and C4). The density of total, slow and fast spindles was calculated as average number of spindles (total, slow and fast) over time spent in N2 and N3.

Spectral power in all EEG frequency bands was also analyzed through the Polysmith software package (Nihon Kohden Polysmith version 9.0). In detail, power spectral analysis was carried using the fast Fourier transform (FFT) technique on all recorded artifact-free epochs of each experimental night, from frontal, left and right (F3, F4) and central, and left and right (C3, C4) EEG derivations. The analysis was run using the Polysmith software, which applies FTT on consecutive 5 s epochs, and then averages the spectral power (units: pico-Watt) of these segments in 30 s epochs. Before analysis, the software converts the data in 250 Hz. We focused our analysis on the following spectral bands: delta 0.5–4 Hz, theta 4–8 Hz, alpha 8–11 Hz, sigma 11–16 Hz and beta 16–32 Hz.

### 2.4. Data Analysis

Statistical analyses were carried out using SPSS statistics software (version 21.0). All PSG variables of the IN and GS participants were averaged across the two nights of recording. The Shapiro–Wilk test showed that variables were normally distributed. However, since the data of the PS group (also averaged across two nights of PSG recording) were not normally distributed [16], Kruskal–Wallis one-way ANOVA was used to compare traditional sleep parameters as well as sleep continuity, stability and organization variables between the three groups. Pairwise comparisons were performed through the Dwass–Steel–Critchlow–Fligner test. Student’s t test was employed, instead, to assess differences between IN and GS participants in microarousal index, spindle and spectral power measures, which were not assessed in the PS group [16]. 

Significance was set at *p* ≤ 0.05. 

## 3. Results

### 3.1. Traditional Sleep Architecture Variables

Table 2 displays between-group comparisons in traditional sleep variables. The model was significant for SOL, WASO% and SE%, and there was a trend to significance for N3%. SOL was significantly higher in the IN compared to the GS group (*p* = 0.004) and it showed a trend to a significant difference between the IN and PS as well (*p* = 0.066). WASO% was higher in the IN relative to the GS group (*p* = 0.018). IN participants’ SE% was significantly lower than that of both PS (*p* < 0.001) and GS subjects (*p* < 0.001). Finally, we found a trend to a significantly higher N3% in IN relative to PS participants (W = 3.15, *p* = 0.067). 

### 3.2. Sleep Continuity

Most sleep continuity measures were significantly impaired both in the PS and the IN group compared to the GS. Significant between-group differences emerged for total (GS: 1.66 ± 1.16, PS: 2.79 ± 1.13, IN: 3.02 ± 1.12; χ^2^ = 15.928; *p* < 0.001; ε^2^ = 0.231) and brief awakenings frequency (GS: 1.46 ± 1.25, PS: 2.62 ± 1.10, IN: 2.56 ± 1.09; χ^2^ = 12.434; *p* = 0.002; ε^2^ = 0.180), with higher frequency in PS and IN participants compared to the GS. Long awakening frequencies showed a trend towards significant differences across groups (GS: 0.191 ± 0.200, PS: 0.170 ± 0.139, IN: 0.487 ± 0.497; χ^2^ = 4.588; *p* = 0.106; χ^2^ = 0.072). These comparisons are displayed in Figure 2. 

Table 3 shows between-group differences in the frequency of awakenings from specific sleep states. The frequency of awakenings from N2 and from N3 differed significantly across groups, with higher frequency in the PS compared to the GS group (from N2: *p* = 0.021; from N3: *p* = 0.04) and in the IN compared to the GS group (from N2: *p* = 0.001; from N3: *p* < 0.001). There was also a trend towards significant differences across groups for the frequency of awakenings from N1, whereas those from REM sleep displayed no differences.

### 3.3. Sleep Stability

All main sleep stability parameters were significantly impaired both in the PS and the IN group compared to the GS (all *p*’s < 0.001): arousal frequency (GS: 4.78 ± 2.63, PS: 7.92 ± 1.95, IN: 6.97 ± 2.45; χ^2^ = 16.423; ε^2^ = 0.238), state transitions frequency (GS: 15 ± 7.64, PS: 23.4 ± 5.52, IN: 22.7 ± 6.06; χ^2^ = 17.461; ε^2^ = 0.253), FU periods frequency (GS: 1.12 ± 0.95, PS: 2.18 ± 0.83, IN: 1.92 ± 0.72; χ^2^ = 16.242; ε^2^ = 0.235) and TFU% (GS: 8.87 ± 7.64, PS: 17.3 ± 6.63, IN: 16.1 ± 7.13; χ^2^ = 16.184; ε^2^ = 0.234) (Figure 3). 

As displayed in Table 4, the frequency of arousals from all sleep states differed across groups. Arousals from N2 to N1, from N3 to N2 and from REM to N1 were more frequent in PS relative to GS participants (*p* = 0.004, *p* = 0.001 and *p* = 0.05, respectively). Moreover, arousals from N3 to N2 tended to be more frequent in IN compared to PS subjects (*p* = 0.102) and those from REM to N1 tended to be more frequent in IN relative to GS participants (*p* = 0.086). Instead, arousals from N3 to N1 were significantly more frequent in the IN compared to the PS group (*p* = 0.016). 

### 3.4. Sleep Cyclic Organization

Significant differences emerged for all three sleep organization measures (Figure 4): number of sleep cycles (GS: 3.93 ± 1.16, PS: 4.80 ± 1.91, IN: 3.24 ± 1.56; χ^2^ = 7.414; *p* = 0.025; ε^2^ = 0.145), TCT% (GS: 60 ± 17.5, PS: 58.3 ± 17.2, IN: 36.7 ± 29.7; χ^2^ = 7.682; *p* = 0.021; ε^2^ = 0.151) and cycle mean duration (GS: 69.3 ± 15.9, PS: 66.6 ± 16.8, IN: 52.1 ± 18.6; χ^2^ = 5.908; *p* = 0.050; ε^2^ = 0.116). 

### 3.5. Sleep Spindles and Microarousals

No between-group differences emerged for the density of total spindles (IN: 2.52 ± 2.37 vs. GS: 1.60 ± 1.09; t(29)= −1.34, *p* = 0.192, Cohen’s d = −0.482), slow spindles (IN: 0.91 ± 0.88 vs. GS: 0.69 ± 0.51; t(29)= −0.82, *p* = 0.416, Cohen’s d = −0.298) and fast spindles (IN: 2.21 ± 2.14 vs. GS: 1.37 ± 0.90; t(29)= −1.35, *p* = 0.185, Cohen’s d = −0.490). The microarousal index, instead, was significantly higher in the IN group (IN: 11.44 ± 4.21 vs. GS: 8.97 ± 2.20; t(29)= −2.04, *p* = 0.05, Cohen’s d = −0.723).

### 3.6. Spectral Power

Between-group differences in spectral power are displayed in Table 5. Significant differences emerged only at C4 in the alpha, sigma and beta bands, with IN participants showing higher spectral power in all three bands. Moreover, the same group showed a trend to significantly higher power in the delta and theta bands, again in the C4 derivation. 

## 4. Discussion

In this study we examined, in a sample of insomniac participants, a set of PSG measures indexing sleep continuity, stability and cyclic organization that had not been previously assessed in this population. These measures were compared to those of a sample of good sleepers and one of healthy individuals with complaints of poor sleep (i.e., individuals with poor subjective sleep quality who do not satisfy criteria for an insomnia disorder). 

We observed a major finding regarding the differences we noted in objective sleep quality profiles across the three groups with different sleep quality perception. Specifically, as expected, we observed overall poorer sleep quality in PS and IN participants compared to the GS group. In PS subjects, sleep cyclic organization was preserved and paralleled by a significant disruption of sleep continuity and especially stability measures, whereas the IN group showed relevant impairments in all three sets of measures as well as in some traditional sleep parameters (i.e., sleep latency, sleep efficiency and WASO%).

This pattern of findings is compatible with the hypothesis that individuals with complaints of poor sleep, though not satisfying criteria for an insomnia diagnosis, may be considered on a continuum with insomniacs, as suggested by longitudinal studies showing that the development of an insomnia syndrome is a frequent long-term evolution of poor subjective sleep quality [26,27]. In fact, we observed a higher degree of impairment of several sleep quality measures in IN relative to PS participants. First of all, the traditional macroscopic measures of sleep fragmentation (e.g., WASO proportion, sleep efficiency) displayed clear alterations in the IN but not in the PS subjects (in line with our previous work showing that traditional sleep parameters did not differ between good and poor sleepers [16]). Furthermore, only IN participants showed significantly longer sleep onset latency relative to GS, and the differences with the PS group in this measure also tended to significance. In addition, sleep cyclic organization was significantly disrupted in the IN but not the PS participants, with fewer sleep cycles and a lower proportion of time spent in sleep cycles in the former group (only the average duration of cycles was similar between the two groups). 

As for our sleep continuity and stability measures, they appear instead mostly comparable between IN and PS subjects. However, a closer look at descriptive data and at the results of pairwise comparisons reveals that the PS group appears slightly more impaired in sleep stability whereas IN subjects appear relatively more impaired in continuity parameters, in line with the more relevant sleep fragmentation displayed by the IN group at the macrostructural level. Thus, it could be hypothesized that, in the IN group, sleep state transitions more frequently result in awakenings. In other words, the force of arousing events would be weaker in the PS compared to the IN group, translating, in the former group, into frequent transitions to shallower sleep states or very brief awakenings but not into the full awakenings which characterize, instead, IN participants’ sleep. Of note, arousals from N3 to N1 were higher in the IN compared to the PS group, as was the frequency of long awakenings (though this difference failed to reach significance). 

This interpretation is coherent with Baekeland and Hoy’s finding [43] that individuals accurately estimate the number of long night awakenings, while being unaware of the briefer ones. Therefore, it could be that a high sleep instability, apparent on a more microstructural level, and accompanied by preserved overall sleep architecture, predisposes individuals to a general perception of poor sleep, but it is not sufficient to determine the sleep complaints that warrant an insomnia diagnosis. In this latter case, a conscious perception of frequent and long sleep interruptions, paralleled by long sleep latency, could be necessary. 

It must be mentioned, however, that our overall pattern of findings is also compatible with another explanation, which considers healthy poor sleepers and insomniacs as separate entities rather than the expression of different degrees of severity on a continuum of non-organic sleep disturbance. This alternative hypothesis is in line with the recent trend of insomnia research to classify individuals diagnosed with insomnia in different phenotypes. For instance, several studies distinguish insomnia with short sleep duration from insomnia with normal sleep duration (also called “sleep state misperception” or “paradoxical insomnia”) (e.g., [44]). Other more refined distinctions refer to sleep onset insomnia [45], sleep maintenance insomnia [45,46], insomnia with early morning awakening [46] and combined phenotypes [45]. In this perspective, our IN participants could be classified as a combined phenotype, based on their significantly longer sleep latency and impaired sleep continuity compared to good sleepers. Therefore, any hypothesis on this data should be extended only to individuals presenting a similar phenotype. 

Furthermore, we are not able to compare our data on IN participants’ sleep instability and disorganization to the previous literature. In fact, to our knowledge, no other study has assessed, in insomniacs, sleep stability measures as defined here (including sleep state transitions, which are instead increasingly used in sleep studies on other populations and topics). Similarly, while objective sleep fragmentation is widely described in insomnia (see [47] for a review], sleep organization has been totally neglected in this population, in line with the general tendency of sleep research to overlook dynamic information on the sleep episode (already underlined by several authors, e.g., [48,49]). Our findings underline the importance of collecting this type of data especially in sleep-disordered populations. In fact, several observations suggest that the NREM-REM cycle subserves important biological functions. For instance, all mammalian species are characterized by a regular alternation of NREM and REM [50]; this alternation survives in basically all pathological conditions, including severe neurological diseases [51]. Moreover, the regular build-up of sleep cycles appears to be essential for basic biological processes, such as protein synthesis and anabolism [52], and it has been recently proposed that the “recovery” function of sleep relies on the NREM-REM alternation rather than merely on slow wave activity [53]. Furthermore, in the frame of the “sequential hypotheses” on sleep-related memory consolidation (reviewed in [54]), it has been proposed that complete and undisturbed sleep cycling plays a relevant role in the efficient processing of newly acquired memories during sleep [55], an idea supported by both correlational [56,57] and experimental data [17,58,59]. In light of this literature, our results suggest that the disturbed dynamics of sleep cycling in insomnia could contribute to the cognitive deficits [60], and especially the impairments in sleep-related memory consolidation [61] documented in this population. 

The finding of disrupted sleep cycling in the IN group is also compatible with the hypothesis, advanced by several authors [21,22], that sleep disturbances that are not accessible to conscious awareness-determined poor sleep perception. Although, based on our data, we cannot exclude that the conscious perception of awakenings is necessary to determine insomniac’s reports of poor sleep (since the IN group presented a clearly disturbed sleep maintenance), more subtle sleep disruptions might still have contributed to their subjective judgments. This idea is also supported by our results on microarousals and spectral power. Microarousal index was significantly higher in the IN compared to the GS group (note that this variable indicates abrupt changes in the EEG lasting less than 15 s, i.e., a shorter time scale than our sleep stability measures, which are defined on a 30 s epoch basis); the former group also showed, in the C4 derivation, enhanced power in the beta, alpha and sigma bands and a trend to higher power in the theta band. Our finding on microarousals is consistent with data from the two other studies addressing this parameter in insomniacs compared to good sleepers [24,62]. Similarly, a more elevated power in insomniacs in the fast frequency bands is already well-documented (see [7,24] for reviews). Enhanced power in the slower bands, instead, is reported much less frequently (e.g., [63,64]). As for the interesting asymmetry we observed in spectral power, it is in line with data from the few previous studies assessing inter- and intra-hemispheric asymmetries in insomniacs’ sleep [64,65,66] and suggesting that insomnia sufferers present specific patterns of EEG power asymmetry which are partially related to their clinical symptoms [66]. 

Finally, no significant differences emerged between IN and GS participants in sleep spindles, which have been implied both in memory consolidation processes (e.g., [67]) and in the protection of sleep maintenance from disturbances linked to external stimuli (e.g., [68]). Therefore, it appears that the spindle-mediated sleep-protective mechanisms, though unaltered compared to the control group, were not able to shield our IN participants’ sleep from significant fragmentation. This evidence is in contrast with the findings by Besset et al. of decreased spindling in insomniacs versus good sleepers [69] and with the study by Andrillon et al. [25], reporting increased spindle frequency in insomniacs with and without sleep misperception relative to good sleepers. However, it is in line with what was observed by Bastien et al. [70], who did not find any between-group differences. In sum, given the inconclusiveness and paucity of studies on the topic, drawing conclusions on spindling in insomniacs appears premature and requires further research. 

A few limitations must be acknowledged which impose caution in the interpretation and generalization of the study’s findings. First, the IN and GS group could not be compared to the PS in terms of circadian preference, habitual sleepiness and anxiety and depression levels, since these data were missing for the latter group [16]. Moreover, as mentioned above, we included all individuals diagnosed with insomnia in the IN group without distinctions based on insomnia subtypes (e.g., insomnia with or without sleep state misperception, etc.). However, based on extant literature [44,45,46], our IN participants would probably be classified as insomniacs without sleep state misperception and with a combined phenotype (i.e., with impairments of sleep onset and sleep maintenance). Therefore, interpretations of our data could be extended to individuals with similar phenotypes. 

## 5. Conclusions

In conclusion, this is the first study to provide an empirical description of a set of PSG-defined sleep measures indexing sleep continuity, stability and cyclic organization in insomniacs relative to good sleepers and to self-reported healthy poor sleepers. Compared to both control groups, insomniacs showed altered sleep continuity (mainly relying on long rather than brief awakenings) and a marked disruption of sleep cycles. Moreover, in line with previous literature, they displayed altered power in several EEG frequency bands and increased microarousal frequency. These findings support the idea that disturbances at various microstructural levels, often inaccessible to conscious awareness, may contribute to poor sleep perception in individuals diagnosed with insomnia [21,22]. Moreover, they add to previous literature attempting to define objective, PSG-based, criteria sets for the description of insomnia [3] by highlighting the role that altered sleep cycling may play in this sleep disorder. Finally, considering that sleep continuity, stability and organization have been implicated in sleep-related learning processes [17,58,59,71], our data suggest that these kinds of impairments should be considered in the explanation of cognitive and, specifically, memory deficits observed in this population. 

## Figures and Tables

**Figure 1 ijerph-20-01240-f001:**
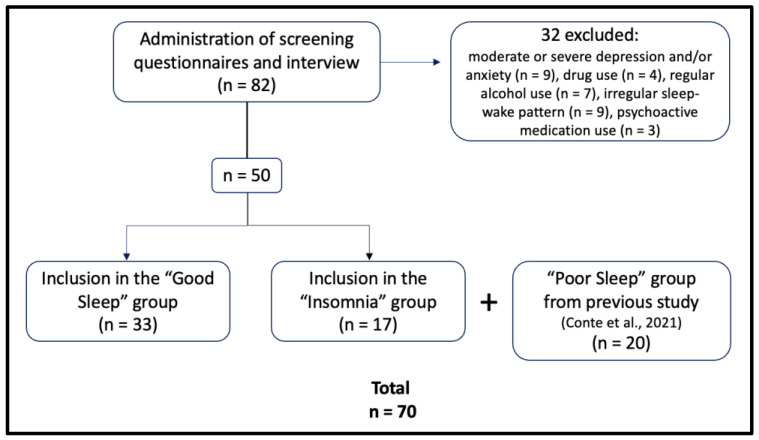
Flowchart of the recruitment process. Data on participants of the “Poor Sleep” group analyzed here come from a previous study of our group [16].

**Figure 2 ijerph-20-01240-f002:**
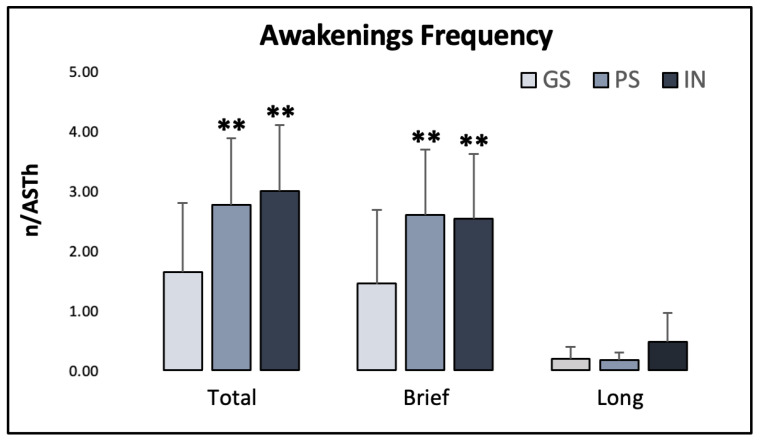
Frequency of total, brief and long awakenings in the three groups. AST: actual sleep time. Asterisks indicate significant differences relative to the GS group (**: *p* ≤ 0.01). Error bars represent standard deviation.

**Figure 3 ijerph-20-01240-f003:**
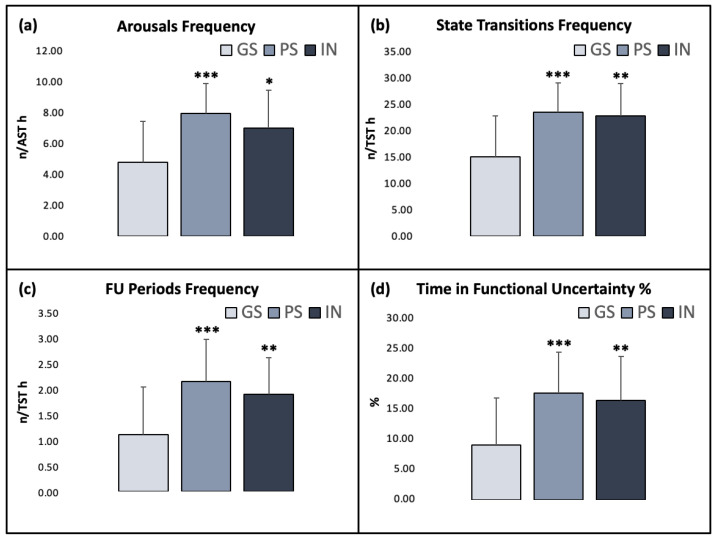
Sleep stability measures in the three groups. (**a**) Frequency of arousals over actual sleep time (AST); (**b**) frequency of state transitions over total sleep time (TST); (**c**) frequency of functional uncertainty (FU) periods over TST; (**d**) time in functional uncertainty (TFU) %. Asterisks indicate significant differences relative to the GS group (*: *p* ≤ 0.05; **: *p* ≤ 0.01; ***: *p* ≤ 0.001). Error bars represent standard deviation.

**Figure 4 ijerph-20-01240-f004:**
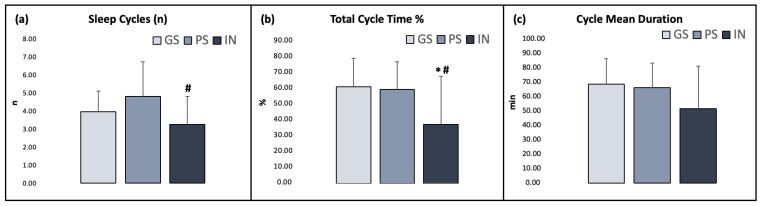
Sleep organization measures in the three groups. (**a**) Number of sleep cycles; (**b**) total cycle time (TCT) %; (**c**) cycle mean duration. Asterisks indicate significant differences relative to the GS group (*: *p* ≤ 0.05). Hashtags indicate significant differences relative to the PS group (#: *p* ≤ 0.05). Error bars represent standard deviation.

**Table 1 ijerph-20-01240-t001:** Age and gender distribution in the three groups.

	GS Group	IN Group	PS Group	Statistic	*p*
Age	27.3 ± 6.18	26.6 ± 6.71	24.5 ± 2.34	t = 0.29	0.074
Gender	12M, 21F	4M, 13F	4M, 16F	χ^2^ = 1.12	0.389

Notes. Data on age are presented as mean ± standard deviation. Age was compared through Student’s *t* test, whereas the results of the chi-squared test are reported for differences in gender distribution.

**Table 2 ijerph-20-01240-t002:** Traditional sleep architecture parameters in the three groups.

	GS	PS	IN	χ^2^	*p*	ε^2^
TIB (min)	465 ± 59.8	465 ± 70.5	491 ± 49.2	1.884	0.390	0.02731
TST (min)	454 ± 61.1	454 ± 69.8	494 ± 149	0.335	0.846	0.00485
AST (min)	438 ± 58.6	437 ± 69.9	425 ± 55.9	0.586	0.746	0.00849
SOL (min)	9.06 ± 7.51	11.6 ± 11.2	17.4 ± 10.3	10.681	**0.005**	0.15707
N1 (%)	9.36 ± 4.47	11.0 ± 4.75	9.90 ± 3.79	2.325	0.313	0.03369
N2 (%)	48.3 ± 9.74	48.4 ± 7.68	44.9 ± 5.81	1.352	0.509	0.01960
N3 (%)	22.6 ± 8.98	19.0 ± 6.22	25.6 ± 11.1	5.645	*0.060*	0.08181
REM (%)	19.7 ± 5.78	21.5 ± 5.86	18.0 ± 5.09	3.957	0.168	0.05176
WASO (%)	3.39 ± 2.81	3.89 ± 2.14	7.91 ± 8.24	7.075	**0.029**	0.10253
SE (%)	94.3 ± 3.11	93.8 ± 3.39	86.5 ± 7.29	24.790	**<0.001**	0.35927

Notes. Data are presented as mean ± standard deviation. Significant *p*-values are in bold. Trends to significance are in italics.

**Table 3 ijerph-20-01240-t003:** Frequency of awakenings from specific sleep states in the three groups.

	GS	PS	IN	χ^2^	*p*	ε^2^
from N1	0.092 ± 0.089	0.126 ± 0.066	0.129 ± 0.069	4.445	*0.108*	0.06442
from N2	0.024 ± 0.019	0.045 ± 0.026	0.054 ± 0.025	15.164	**<0.001**	0.21976
from N3	0.014 ± 0.015	0.026 ± 0.016	0.271 ± 0.961	16.051	**<0.001**	0.23263
from REM	0.031 ± 0.028	0.037 ± 0.029	0.044 ± 0.039	1.416	0.493	0.02053

Notes. Data are presented as mean ± standard deviation. Significant *p*-values are in bold. Trends to significance are in italics. Awakenings from a certain sleep stage are calculated as frequencies over the total time spent in that stage (minutes).

**Table 4 ijerph-20-01240-t004:** Frequency of arousals from specific sleep states in the three groups.

	GS	PS	IN	χ^2^	*p*	ε^2^
N2 to N1	0.065 ± 0.040	0.111 ± 0.049	0.099 ± 0.058	11.184	**0.004**	0.162
N3 to N1	0.013 ± 0.017	0.007 ± 0.011	0.197 ± 0.723	7.759	**0.021**	0.112
N3 to N2	0.185 ± 0.337	0.269 ± 0.161	0.740 ± 2.39	13.568	**0.001**	0.197
REM to N1	0.096 ± 0.087	0.148 ± 0.072	0.141 ± 0.068	7.455	**0.024**	0.108

Notes. Data are presented as mean ± standard deviation. Significant *p* values are in bold. Arousals from a certain sleep stage are calculated as frequencies over the total time spent in that stage (minutes).

**Table 5 ijerph-20-01240-t005:** Spectral power in the delta, theta, alpha, sigma and beta bands in the IN and GS groups.

		IN Group	GS Group	t	*p*	Cohen’s d
DELTA	F3	446.53 ± 201.34	489.30 ± 293.23	0.48	0.631	0.172
	F4	371.28 ± 268.46	364.58 ± 258.60	−0.06	0.946	−0.024
	C3	391.41 ± 183.98	430.05 ± 244.18	0.51	0.614	0.180
	C4	374.05 ± 159.50	282.50 ± 156.79	−10.63	0.113	−0.578
	O1	687.85 ± 1173.64	425.19 ± 315.49	−0.84	0.408	−0.297
	O2	678.14 ± 1181.01	456.79 ± 313.65	−0.70	0.487	−0.249
THETA	F3	89.55 ± 44.59	96.10 ± 84.25	0.27	0.782	0.099
	F4	74.44 ± 43.58	69.93 ± 76.23	−0.20	0.836	−0.073
	C3	72.69 ± 35.45	69.71 ± 53.41	−0.18	0.852	−0.066
	C4	76.43 ± 45.13	49.68 ± 29.50	−10.95	*0.060*	−0.692
	O1	237.29 ± 764.45	68.62 ± 92.93	−0.87	0.393	−0.307
	O2	241.43 ± 754.68	75.15 ± 87.56	−0.86	0.398	−0.303
ALPHA	F3	14.92 ± 5.31	15.44 ± 15.34	0.13	0.897	0.046
	F4	12.19 ± 8.65	10.19 ± 13.14	−0.51	0.613	−0.181
	C3	11.88 ± 4.87	10.92 ± 9.55	−0.37	0.716	−0.123
	C4	13.63 ± 5.90	8.22 ± 6.74	−20.42	**0.022**	−0.859
	O1	40.04 ± 108.97	12.32 ± 21.14	−0.97	0.341	−0.342
	O2	39.08 ± 109.08	12.55 ± 18.35	−0.93	0.361	−0.329
SIGMA	F3	8.48 ± 3.52	7.21 ± 7.45	−0.62	0.536	−0.220
	F4	6.75 ± 5.14	4.73 ± 6.26	−10.01	0.323	−0.356
	C3	7.90 ± 4.18	5.97 ± 4.27	−10.29	0.207	−0.457
	C4	8.95 ± 4.89	4.57 ± 3.33	−20.92	**0.007**	−10.047
	O1	19.92 ± 60.84	4.93 ± 8.17	−0.95	0.352	−0.335
	O2	20.16 ± 60.76	5.52 ± 7.52	−0.93	0.362	−0.328
BETA	F3	8.70 ± 3.42	6.80 ± 7.96	−0.89	0.378	−0.317
	F4	6.31 ± 3.86	5.04 ± 6.81	−0.65	0.515	−0.233
	C3	7.96 ± 4.49	6.25 ± 5.40	−0.98	0.336	−0.346
	C4	8.54 ± 6.03	4.24 ± 2.60	−20.55	**0.016**	−0.905
	O1	21.65 ± 67.20	4.76 ± 8.62	−0.96	0.343	−0.341
	O2	21.49 ± 67.16	5.78 ± 8.35	−0.89	0.376	−0.318

Notes. Data are presented as mean ± standard deviation. Significant *p*-values are in bold. Trends to significance are in italics.

## Data Availability

The data presented in this study are available on request from the corresponding author. The data are not publicly available due to privacy reasons.

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
