# Peer review of "Sleep Continuity, Stability and Cyclic Organization Are Impaired in Insomniacs: A Case–Control Study"

_ijerph, 2023, doi:10.3390/ijerph20021240_

Round 1
Reviewer 1 Report
Need demographic clarification on differences or similarities between PS and GS or IN groups. More clarification regarding why PS are deemed so. Explain criteria for this.
Author Response
Need demographic clarification on differences or similarities between PS and GS or IN groups. More clarification regarding why PS are deemed so. Explain criteria for this.
We thank the Reviewer for this comment. Following also Reviewer 2’s suggestion, we have added the PS group’s data in Table 1, showing that the 3 groups did not differ in terms of demographic factors. Moreover, the criteria for the recruitment of PS are now more clearly specified at lines 179-185.
(0) According to reviewer #1 suggested to Assistant Editor, please delete references:
"There were several papers which contained one or more authors as references
may have been added unnecessarily and other research papers could have been
cited as examples. 18,19, 52,53, 56,57,58 and 6"
(I) Please check that all references are relevant to the contents of the
manuscript.
References 19, 52, 57 have been deleted or substituted. Specifically, reference 19 (Cellini et al., 2021) has been substituted with Casagrande et al., 2020; reference 52 (Conte and Ficca, 2013), now 54, has been substituted with Sara, 2017; reference 57 (Conte et al., 2012) has been deleted.
Reviewer 2 Report
Dear authors,
Thank you for the very interesting paper. I have some questions below:
1. Were there any difference between the 3 groups in terms of demographic data (i,e age, sex, etc..). The average data was shown, but no statistical comparison was in the table.
2. There is a typo error on line 311" Table...."
3. Do you have any explanation for the difference of spectral analysis only on the C4 and not C3?
Author Response
1. Were there any difference between the 3 groups in terms of demographic data (i,e age, sex, etc..). The average data was shown, but no statistical comparison was in the table.
We thank the Reviewer for his comment. Following also Reviewer 1’ s suggestion, we have added the PS group’s data in Table 1, showing that the 3 groups did not differ in terms of demographic factors.
2. There is a typo error on line 311" Table...."
We thank the Reviewer for noticing the typo, which is now corrected.
3. Do you have any explanation for the difference of spectral analysis only on the C4 and not C3?
This is an interesting point. We do not have any specific hypothesis on this asymmetry, though we did review previous literature on this issue. Actually, only three studies (Kovrov et al., 2006; St Jean et al., 2012; Provencher et al., 2020) have addressed and found EEG topographic asymmetries in insomniacs’ sleep (one of which specifically observed C3-C4 asymmetry as in our study, Kovrov et al., 2006). Their findings point to a high variability of this asymmetry within the night. Unfortunately, variations of the EEG spectral power across different sleep states were not assessed in our study. Moreover, two of the studies (St Jean et al., 2012; Provencher et al., 2020) based their working hypotheses on the existence of differences regarding insomniacs with sleep-state misperception relative to insomniacs with no misperception, a distinction that was not made in our sample. Anyway, neither of the extant three studies produced specific hypotheses, compatible with our data, on the meaning of the asymmetries (except for the general hypothesis that insomniacs present EEG asymmetries linked to their clinical manifestations). We have added a comment on this interesting issue in the Discussion and cited the relevant studies.